# Hydrogenation of iron in the early stage of Earth's evolution

Riko Iizuka-Oku[1,2], Takehiko Yagi[2], Hirotada Gotou[3], Takuo Okuchi[4], Takanori Hattori[5] & Asami Sano-Furukawa[5]

Density of the Earth's core is lower than that of pure iron and the light element(s) in the core is a long-standing problem. Hydrogen is the most abundant element in the solar system and thus one of the important candidates. However, the dissolution process of hydrogen into iron remained unclear. Here we carry out high-pressure and high-temperature *in situ* neutron diffraction experiments and clarify that when the mixture of iron and hydrous minerals are heated, iron is hydrogenized soon after the hydrous mineral is dehydrated. This implies that early in the Earth's evolution, as the accumulated primordial material became hotter, the dissolution of hydrogen into iron occurred before any other materials melted. This suggests that hydrogen is likely the first light element dissolved into iron during the Earth's evolution and it may affect the behaviour of the other light elements in the later processes.

[1] Geodynamics Research Center, Ehime University, Matsuyama, Ehime 790-8577, Japan. [2] Geochemical Research Center, Graduate School of Science, The University of Tokyo, Bunkyo-ku, Tokyo 113-0033, Japan. [3] Institute for Solid State Physics, The University of Tokyo, Kashiwa, Chiba 277-8581, Japan. [4] Institute for Planetary Materials, Okayama University, Misasa, Tottori 682-0193, Japan. [5] Japan Proton Accelerator Research Complex (J-PARC) Center, Japan Atomic Energy Agency, Tokai, Naka, Ibaraki 319-1195, Japan. Correspondence and requests for materials should be addressed to R.I.-O. (email: riizuka@eqchem.s.u-tokyo.ac.jp) or to T.Y. (email: yagi@eqchem.s.u-tokyo.ac.jp).

Since the Earth's core was first clarified to be less dense than pure iron[1], numerous studies have investigated which light element(s) might be present in the core, including silicon, sulfur, oxygen, hydrogen and carbon. Experimental studies have sought to clarify the behaviour of these elements and iron under high-pressure and high-temperature (high-$P$–$T$) conditions[2]. Hydrogen is the most abundant element in the solar system, and was long ago suggested as a possible major light element in Earth's core[3]. The solubility of hydrogen in iron is very low at ambient pressure, but increases greatly at high pressure[4]. Experiments on iron and hydrous minerals[5–8] have shown, albeit indirectly, that hydrogen might indeed be one of the light elements in the core. Other light elements can stably remain in iron once incorporated under high-$P$–$T$ conditions, and thus their behaviour can be studied in detail using recovered samples. However, iron hydride is formed only at high pressures, and hydrogen is released rapidly and completely after the hydride is recovered to ambient conditions. Therefore, the behaviour of hydrogen has been much more difficult to study than the other light elements. Various studies have inferred conclusions based on indirect evidence such as the texture change of the recovered sample[8–10], structural or volume changes of iron under high pressure[4,11–18] or the observation of bubbles in iron recovered by a very rapid pressure decrease[19]. Recently, studies from new approach such as first-principles calculations[20,21] or isotope measurement[22] were reported but the results are still controversial. Here we report the results of the first in situ neutron experiments under high-$P$–$T$ conditions to directly observe the behaviour of hydrogen.

Hydrogen shows little scattering of X-rays. High-$P$–$T$ in situ X-ray observations can provide only the unit cell volume of iron, which increases with the dissolution of hydrogen. The amount of hydrogen in iron hydride has been estimated using the volume–composition relations of various metal hydrides[15,18,23,24]. Neutron scattering has been widely used to elucidate the behaviour of hydrogen, because it provides direct information on hydrogen due to its relatively large scattering power. So far, however, neutron beam fluxes have remained very low, and there has been no in situ observation of iron hydride or deuteride under pressure. Although high-pressure phases of iron hydride and its deuteride that form by the reaction of iron and hydrogen or deuterium have previously been examined by powder neutron diffraction[25], the observations were made at ambient pressure in a metastable condition after quenching by rapid freezing. A new beamline dedicated to high-$P$–$T$ in situ neutron studies was recently constructed at the pulsed spallation neutron source at the Japan Proton Accelerator Research Complex (J-PARC), Tokai, Japan. This new beamline (BL11), named 'PLANET'[26], allows in situ neutron studies up to ~10 GPa at high temperatures, corresponding to the $P$–$T$ conditions in the upper mantle[27].

In this study, we directly observe the behaviour of iron and hydrous minerals to simulate the processes that operate early in the Earth's evolution, and succeed in directly observing that the water released by the dehydration of hydrous minerals reacts with iron to form both iron oxides and iron hydride.

## Results

**High-pressure and -temperature in situ neutron experiments.** A cylindrical iron sample embedded in a 1:1 mixture of $Mg(OD)_2$ and $SiO_2$ was subjected to high-$P$–$T$ conditions using a multi-anvil apparatus with six independently acting 500 ton rams (ATSUHIME 6-axis press)[27] installed in the PLANET beamline[26]. This starting material is a simple representation of the primordial materials of the Earth: iron–enstatite–water. Some experiments were also conducted by replacing $Mg(OD)_2$ with MgO to clarify the role of $D_2O$ (Supplementary Table 1). These samples were placed in a graphite capsule and compressed to ~5 GPa at room temperature (see Methods; Supplementary Figs 1 and 2). The temperature was then increased stepwise while keeping the applied load constant, and neutron diffraction

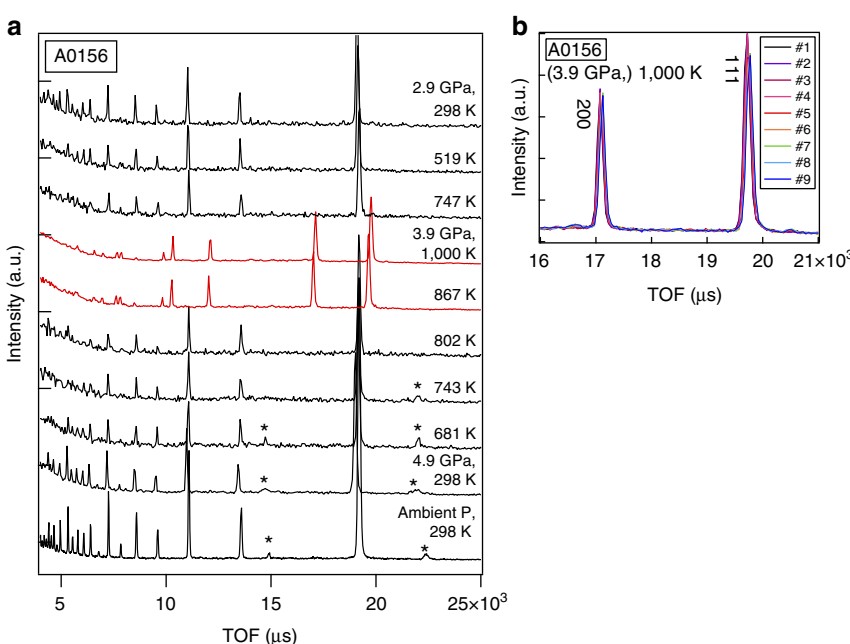

**Figure 1 | Results of powder neutron diffraction.** (**a**) Entire data set of diffraction patterns observed in run #A0156. The observations were made from bottom to top. The red patterns indicate the iron or iron hydride in an fcc structure. Most of the intense lines are from the iron sample, while the two peaks marked with asterisks are from $Mg(OD)_2$, and disappeared upon heating. (**b**) Bragg reflections of the 111 and 200 peaks of fcc iron observed at constant conditions of 3.9 GPa and 1,000 K for ~9 h. Each run (#1–#9) was obtained during exposure for ~1 h. Peak positions shifted slightly with time without changing width.

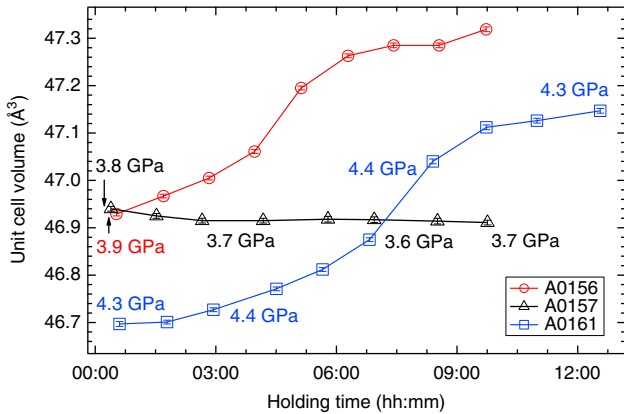

**Figure 2 | Change in the unit cell volume of fcc iron after holding at around 1,000 K for three experimental runs.** #A0156 and #A0161 are for runs with $Mg(OD)_2$, while #A0157 is with MgO in the starting material (details in Supplementary Table 1). Data were collected continuously, and each data point was calculated using data obtained every hour. Errors were estimated based on the standard deviation of Rietveld analysis. Pressures in the figure were calculated from the unit cell volume of NaCl measured during the holding time and the variation was less than ± 0.05 GPa. In run #A0156, the pressure was measured only at the beginning.

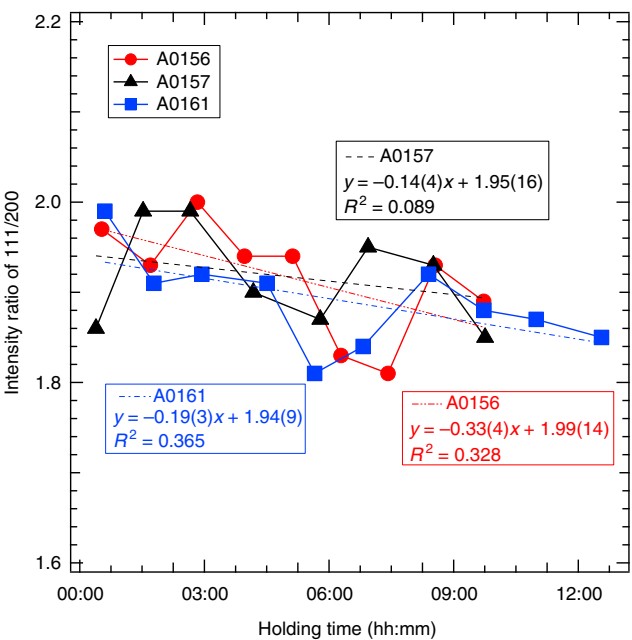

**Figure 3 | Intensity ratio of the 111 and 200 diffraction peaks of fcc iron versus measurement time.** Each data point was calculated based on the diffraction collected for 1 h and liner fittings were made for each run. Although the data are highly scattered owing to the effect of grain growth of the sample, the ratio decreased clearly in runs #A0156 and #A0161. It looks decreased also slightly in #A0157, which was made in dry condition, but the small coefficient of determination ($R^2$) value suggests that this decrease is not meaningful.

patterns were recorded (Fig. 1 and Supplementary Fig. 3). Bragg reflections from $Mg(OD)_2$ were clearly observed before heating, together with those from Fe, but suddenly weakened at 800 K, suggesting the dehydration of $Mg(OD)_2$ to release fluid $D_2O$. When the temperature reached 860 K, the bcc structure of iron transformed to fcc, consistent with the phase diagram of pure iron. The sample was further heated to ~1,000 K, and then held at that temperature for 9–12 h. Neutron diffraction measurements were repeated while the temperature was kept constant. The unit cell volume of fcc iron was obtained as a function of holding time in three independent runs (Fig. 2 and Supplementary Table 2). During these long holding times, the pressures remained almost unchanged. In the two runs #A0161 and #A0156, however, the unit cell volumes of iron increased by 0.96(2)% and 0.83(1)%, respectively. On the other hand, no meaningful change in the unit cell volume was observed in run #A0157, which was kept under dry conditions by replacing $Mg(OD)_2$ in the starting material with MgO. In the neutron diffraction, the relative intensity ratio of the 200 and 111 reflections of fcc iron is sensitive to the amount of hydrogen in the iron (Machida et al.[28]). The calculated ratios of 111/200 versus holding time observed for three runs are plotted in Fig. 3. Because of the grain growth in the sample during the long holding time, the intensity ratio fluctuated a lot and linear fittings were made for each run. In runs #A0156 and #A0161 the ratio decreased at a rate of − 0.33 per hour (the coefficient of determination $R^2 = 0.328$) and − 0.19 per hour ($R^2 = 0.365$), respectively. Although the linear fitting for Run #A0157, also showed a slight decrease at a rate of − 0.14 per hour ($R^2 = 0.089$), $R^2$ is quite small, which indicates no meaningful change was observed with time under dry condition. Although the intensity data are highly scattered, this decrease in the 111/200 ratio, together with the clear increase in the unit cell volume, suggests the increased deuterium content in the iron. Rietveld analysis of the entire diffraction patterns was also made using data accumulated for ~1 h during the 12 h holding time. The results are again widely scattered because of the effect of iron deuteride grain growth, but the data are well explained by the presence of ~0.66(3) and

0.58(2) weight % (wt.%) deuterium in the fcc iron in runs #A0161 and #A0156, respectively (Supplementary Fig. 4 and Supplementary Table 2). These values are in harmony with those obtained by using the volume expansion rate. Deuterium contents calculated from the unit cell volumes at the end of the two runs are 0.79(2) and 0.71(1) wt.%, respectively, by adopting the following relation obtained for a pure Fe–$D_2$ system[28]: $x = \{V(FeD_x) − V(Fe)\}/\Delta V_D$, where $\Delta V_D$ (the volume increase per deuterium atom) is 2.21 Å$^3$.

**Recovered samples.** After holding the sample at 1,000 K (#A0156 and #A0157) or at 956 K (#A0161) for ~12 h, subsequent cooling transformed the fcc iron back into a bcc structure (Fig. 1 and Supplementary Fig. 3). The samples were then recovered at ambient condition and examined by X-ray diffraction. The cross-sections were also observed by scanning electron microscopy (SEM; Fig. 4). The surface of the cylindrical iron sample in run #A0161 was covered with a thin layer of $Fe_{0.96}O$, in harmony with the X-ray observations. The 1:1 powder mixture of $Mg(OD)_2$ and $SiO_2$ surrounding the iron reacted, and changed to olivine $(Mg,Fe)_2SiO_4$ and pyroxene $(Mg,Fe)SiO_3$. The iron content of the olivine (as expressed by the Fe/Mg ratio) was ~0.1–0.7, while that of the pyroxene was 0.01–0.12. Olivines rich in iron were preferentially located near the iron rod, although some also formed farther away than the pyroxene. The iron contained many small vacant holes of a few microns in diameter, but the X-ray diffraction study clarified that the recovered iron has a bcc structure with an identical unit cell to the starting material.

These observations can be accounted for by the following process in the sample chamber. Heating a mixture of Fe,

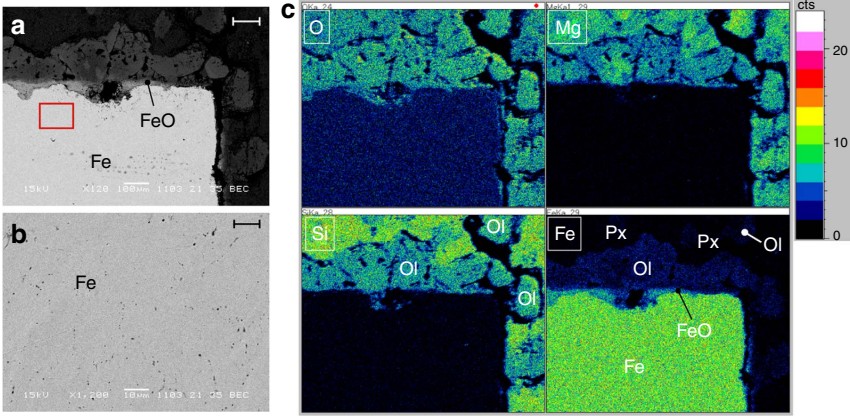

**Figure 4 | Results of SEM analysis on the polished cross-section of the recovered sample.** The sample number is #A0161. (**a**) Representative electron image together with (**b**) enlarged image of the area surrounded by a red square, and (**c**) elemental maps (O, Si, Mg and Fe). The scale bars in **a,b** indicate 100 and 10 μm, respectively. The bright area is metallic iron; a thin FeO layer formed on the surface of the iron in (**a**). Small black or grey dots in iron are vacant holes as clearly shown in **b**. Ol stands for olivine, and Px denotes pyroxene. The surrounding mixture of $Mg(OD)_2$ and $SiO_2$ changed into olivine and pyroxene containing Fe. The former is preferentially located close to the iron in **c**.

$Mg(OD)_2$ and $SiO_2$ at ~5 GPa first led to the decomposition of $Mg(OD)_2$ to MgO and $D_2O$. The $D_2O$ then reacted with iron to form $Fe_{0.96}O$ and $FeD_x$. The decomposed MgO and $SiO_2$ reacted to form $MgSiO_3$ pyroxene, but a large part of the FeO also reacted with MgO and $SiO_2$ to form iron containing olivine and pyroxene. The limited diffusion speed of iron led to iron-rich olivine forming close to the iron sample, while iron-poor pyroxene was dominant far from the iron.

**Deuteration reaction in iron-silicate water system.** The $P$–$T$ paths of the present experiments are plotted in Fig. 5 together with a previously elucidated phase diagram of iron hydride[11,15,18,28,29] and a phase diagram of pure Fe. The amount of deuterium in the iron of the present study (0.71(1)–0.79(2) wt.%) is about one-third of that reported in the pure Fe–$D_2$ system studied recently using the same apparatus[28]. That study found that ~2.24 wt.% deuterium was dissolved in iron at ~6.3 GPa and 988 K, and that the deuteration of iron was completed rapidly within 20 min for a sample of similar size to ours. On the other hand, in our experiments the deuteration of iron occurred gradually over a period of more than 9 h, and the amount of deuterium was smaller even after a much longer holding time. These differences between the studies can be explained by differences in experimental conditions, such as the chemical system, the deuteration reaction speed, the capsule materials and the deuteration pressure. In the study of Fe–$D_2$ system[28], Fe reacted directly with pure $D_2$ that formed by the decomposition of $AlD_3$, while in our experiment $D_2O$ was first released from the hydrous mineral. At low pressures iron is known to dissociate water and the reaction of iron and water results in the formation of iron oxide and hydrogen, or iron hydroxides. Above ~3 GPa, however, $FeH_x$ exists as a stable phase and in the present system both FeO and $FeD_x$ were formed by the following reaction:

$$2Fe + D_2O \rightarrow FeO + FeD_x + (1 - x/2)D_2 \uparrow . \quad (1)$$

This process forms FeO on the surface of the iron sample. Although a large part of FeO reacts with both $SiO_2$ and MgO and forms iron rich silicates, a thin FeO layer (Fig. 4) covers the surface of the iron sample. The diffusivity of deuterium in FeO is poorly known, but the deuteration of iron in the present study was very slow probably due to this FeO layer. The existence of small bubbles in the recovered iron also suggests the low permeability of the FeO layer for deuterium,

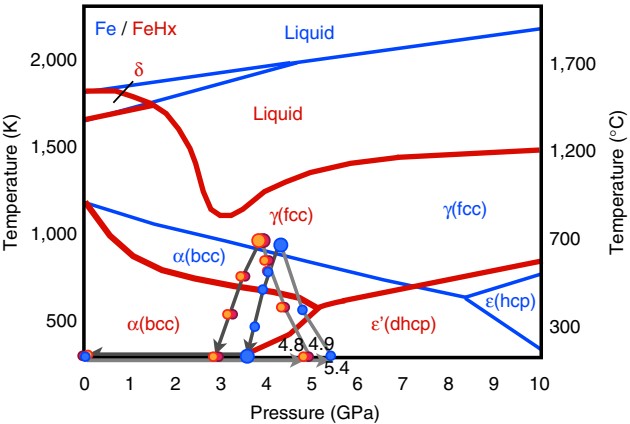

**Figure 5 | Phase diagram of pure Fe (blue lines) and FeH$_x$ (red lines).** Phase boundaries[11,15,18,28,29] and the $P$–$T$ paths of the present three experiments are shown. The small circles represent the $P$–$T$ conditions under which quick observations for 20 min were made, while the large circles represent the $P$–$T$ conditions under which longer measurements (~10 h) were made.

because such bubbles do not usually exist in the iron–hydrous mineral systems when the pressure is released slowly over a few hours. Once the deuterium enters iron, however, its diffusion is sufficiently high to achieve a uniform composition. This is clearly shown by the unchanged half width of the Bragg reflections from iron deuteride, although their positions clearly shifted with time (Fig. 1 and Supplementary Fig. 3). Machida et al.[28] used NaCl as a sample capsule, whereas we used graphite after trying various materials (see Methods). The diffusion of hydrogen in alkali halide is known to be very small, while that in graphite is poorly known. It is possible that a large part of the deuterium that formed by the dissociation of $D_2O$ in our experiments escaped outside the capsule during the long holding time. This possibility is supported by the results of Machida et al.[28] who observed the diffraction of solid $D_2$ when a sample was cooled to room temperature, whereas we did not observe it at all. The lower deuteration pressure in our study (3.9 and 4.3 GPa versus 6.3 GPa in their study) could be an additional factor if the calculated hydrogen contents of $FeH_x$ (refs 30,31) at various $P$–$T$ conditions are considered.

All previous experimental studies of iron and hydrous mineral systems were carried out with heating to much higher temperatures over a short time, and these studies inferred the dissolution of hydrogen into the iron from indirect evidence; however, once the hydrogen was dissolved into iron under high-$P$–$T$ conditions, it was stable and was retained even when the iron hydride was melted[19]. The present study clearly shows that the hydrogenation of iron occurs below 1,000 K when the hydrous mineral is decomposed and water is released into the system. Water also contributed to the formation of FeO and the production of iron-rich olivine.

## Discussion

Many different scenarios have been proposed for the evolution of the Earth, but it is generally assumed that the accretion of primordial material and its gradual heating occurred by the release of gravitational energy. The present study shows that the hydrogenation of iron could have occurred during this very early stage of the Earth's evolution. Since FeH$_x$ is stabilized above $\sim$3 GPa (ref. 4), which is close to the pressure in the centre of the moon, the reaction (1) starts when the mass of primordial material exceeds only $\sim$1% of the present Earth. Moreover, this reaction occurs when iron remains solid. Other light elements are considered to dissolve into molten iron[2] at much higher temperatures. When solid iron is hydrogenated, its melting point drops dramatically, by more than 500 K (Fig. 5). Once the iron hydride melts, it remains much more dense than silicates, and gravitational separation occurs easily. As a result, it is expected that a large part of the hydrogen dissolved in iron at low temperature was carried into the Earth's core. Many Earth's formation models suggest the existence of giant impact after the Earth had reached particular size. Since the hydrogen dissolved in molten iron is retained down to a few GPa (ref. 19), it is likely that a large part of hydrogen carried to the core remained even after the giant impact. The measured density of iron hydride under high-$P$–$T$ conditions suggests that $\sim$1.0 wt.% (ref. 32) or 1.3 wt.% (ref. 24) hydrogen in iron is enough to explain the 10% density deficit of the outer core. It is widely accepted that the primordial material of the Earth (that is, condensates of nebulae formed at various temperatures) contained water. Various calculations suggest that if only 10% of the accreting primordial material was a low-temperature condensate, as per one well-accepted model[33], the hydrogen contained as water in that material would have been sufficient to account for the majority of the density deficit[3,19,34]. The present study showed that water contained in the primordial material reacted at low temperatures to form iron hydride. Although the observed deuterium content of iron is 0.71(1)–0.79(2) wt.%, this low value was probably affected by the escape of deuterium and/or water from the capsule during the experiments. In nature, hydrogen formed by the dehydration of minerals is expected to have high reactivity with iron because of its large-scale uniform distribution. Therefore, it is possible that hydrogen was the primary light element in the core in the very early stage of Earth's history and a large part of that has been retained in the later processes. Recent works studying the p-wave velocity of molten iron at up to 70 GPa (ref. 35) and estimating the low temperature at the top of the core[36] strongly suggest the presence of hydrogen in the core, something to which the present study adds further evidence.

The present results also suggest the importance of studying the effect of hydrogen on the partition coefficient of other light elements into iron. Light elements other than hydrogen must also have been dissolved in iron during the various

processes of the Earth's evolution. Numerous studies have been undertaken under a wide range of $P$–$T$ conditions to clarify the partitioning of various light elements between pure iron and silicates. The present study suggests that it is important to study the partitioning of the light elements between iron hydride and silicates, because the dissolution of hydrogen dramatically changes the properties of iron, as shown by the large reduction in melting point (Fig. 5).

## Methods

**Starting materials.** The starting material comprised an iron rod (99.5%, Nilaco Corp.), SiO$_2$ powder (silicic anhydride precipitated; special grade chemicals, Wako Pure Chemical Industries) and MgO (assay minimum of 98.0%; special grade chemicals, Wako Pure Chemical Industries) or Mg(OD)$_2$ powders. The SiO$_2$ powder was heated at 1,050 °C for $\sim$12 h and then stored in an oven at 110 °C. MgO was heated at 1,100 °C for $\sim$8 h to remove impurities such as absorbed water and MgCO$_3$, and then also stored in an oven at 110 °C. To reduce the high background scattering due to the large incoherent neutron scattering cross-section of hydrogen, deuterated hydroxide Mg(OD)$_2$ was used instead of Mg(OH)$_2$. By comparing with the results of previous studies[9,19], we could not find any difference by using Mg(OD)$_2$, instead of Mg(OH)$_2$. Mg(OD)$_2$ powder was synthesized from this MgO powder and excess D$_2$O (minimum isotope purity of 99.96 atom %D; Aldrich Chemical) in a Teflon-lined stainless steel autoclave at 235 °C for 1 week. After the hydrothermal process was complete, precipitates were filtered and washed with D$_2$O. The obtained white products were dried at room temperature under vacuum. All synthesized samples were well-crystallized powders with no impurities, as inferred from powder X-ray diffraction patterns using an X-ray diffractometer with a position sensitive proportional counter (40 kV, 200 mA, CrKa; Rigaku), and from Raman spectra. High-pressure experiments used powder that had been well ground by a mortar. The iron rod was cut to shape using a cutting machine immediately before sampling to avoid oxidization of its surface.

**High-pressure and high-temperature neutron experiments.** Neutron diffraction patterns were obtained at high-$P$–$T$ using a multi-anvil apparatus with six independently acting 500 ton rams (ATSUHIME 6-axis press)[27] installed in the PLANET beamline (BL11)[26] of the spallation neutron source of the Materials and Life Science Experimental Facility at the J-PARC, Tokai, Japan. A multi-anvil 6-6 type (MA6-6) assembly[37] was used with an improved anvil assembly optimized for the current experiments (Supplementary Fig. 1). The second-stage anvils were made of Ni-binded cylindrical tungsten carbide with truncated edge lengths of 10 mm; a steel supporting jacket was fitted to reduce the risk of blowouts. A guide frame made of Al-alloy with slits for incident and diffracted neutrons was assembled together with the six second-stage anvils and sample cells; it was covered with 0.5-mm-thick glass–epoxy seats to avoid leakage of the neutron-activated samples.

The cell assembly is shown in Supplementary Fig. 2. The pressure medium consisted of a ZrO$_2$ cube with an edge length of 15 mm. A graphite cylinder and disks were used as a furnace, and the electrodes were molybdenum foils. Preformed gaskets made of pyrophyllite, which were fired in advance at 700 °C for 15 min, were pasted on the slope face of the second-stage anvils.

Graphite sample capsules were used instead of NaCl, because D$_2$O—which dissolves NaCl—was formed during the reaction. Preliminary experiments had tested many other materials for the capsule such as Ta, Pt, Fe, Re, Si$_3$N$_4$, AgPd, $h$BN, Al$_2$O$_3$ and MgO, which are conventionally used for the high-$P$–$T$ experiments. Even though most of these materials have high melting points and/or high hardness, the presence of hydrogen considerably altered either the melting temperature or the reaction properties of these materials. We had great difficulties in finding an appropriate material for neutron experiments, which satisfies both low neutron absorption and low reactivity in water containing system. Graphite was finally selected as the best material for the present study.

An iron rod ($\phi$ 2.3 mm, height 1.7 mm, $\sim$60 mg) was carefully placed in the middle of the graphite sample capsule and surrounded by SiO$_2$–Mg(OD)$_2$ powder. The molar ratio of the sample and powder was Fe:SiO$_2$:Mg(OD)$_2$ = 2:1:1. The incident neutron beam was truncated by a slit to be 1 mm × 2 mm before its introduction to the sample. The pressures were calculated using equations of states for Fe (at room temperature)[38] and NaCl (ref. 39) pellets placed above and below the sample. The variation in pressure values was less than ± 0.05 GPa. Temperature calibrations were conducted in separate experimental runs using thermocouple and the temperature fluctuation was within 3 K during the long holding time. Three experiments were carried out as summarized in Supplementary Table 1.

Representative diffraction patterns were obtained in experimental runs #A0157 and #A0161 (Supplementary Fig. 3; see also Fig. 1 for #A0156). The exposure time for each pattern was $\sim$20 min. Although the sample was surrounded by various materials such as the graphite capsule, the heater and the solid pressure-transmitting medium (Supplementary Fig. 2), almost pure diffraction patterns from only the sample were obtained by the use of radial collimators installed

between the sample and the detectors. The molar ratio of the iron and the surrounding oxide mixture was almost 1:1, but most of the strong Bragg peaks were derived from the iron placed at the centre of the sample chamber. Bragg reflections from $Mg(OD)_2$ were also clearly observed before heating. The pressure was increased to $\sim 5$ GPa at room temperature in all these runs, but dropped by $\sim 1$ GPa when the temperature was increased to $\sim 1,000$ K owing to the flow of the solid $ZrO_2$ pressure-transmitting medium and the pyrophyllite gaskets, and also to the phase transition of the sample. Data were collected every hour during measurement for $\sim 10$ h at the maximum temperature of $\sim 1,000$ K. Detailed analyses of the data (volume change, expansion, and deuteration during the holding time) are listed in Supplementary Table 2.

Rietveld analysis requires correction of the intensity observed under high-$P$–$T$ conditions. Separate datasets were collected for an empty cell and for one containing a vanadium pellet at ambient conditions by placing them in the high-pressure cell. The size of each cell assembly was adjusted, so that the neutron pass became identical to that during sample measurement at high pressures. Sample data were normalized with the vanadium data to correct the energy profile of the incident neutron beam, detector sensitivity, and cell attenuation. The empty-cell data were subtracted from the individual data. The initial structure model for fcc iron hydride was taken from Machida et al.[28] with deuterium atoms occupying both the T and O sites. All the data were listed in Supplementary Table 2.

**SEM analysis.** The recovered sample was cut in half, and the surface was carefully polished and coated with carbon by several nm thick. The microstructure was observed by SEM (JSM-5600, JEOL; electron energy of 15 kV) at the Institute for Solid State Physics, the University of Tokyo, to identify the products, surface textures, and chemical distribution in the sample. Chemical composition and elemental mapping data were also obtained by a combination of SEM and energy dispersive X-ray microanalyses. The measurement time for one spectrum was 90 s. The mapping data were collected for $\sim 3$ h.

**Synchrotron X-ray diffraction analysis.** After the SEM analyses, X-ray diffraction measurements of the same recovered samples were conducted using a synchrotron X-ray beam at the AR-NE7 beamline, the photon factory, High Energy Accelerator Research Organization (KEK), Tsukuba, Japan. A white X-ray beam (0.1 mm height, 0.5 mm width) was directed horizontally at the sample through the gasket. Diffracted X-rays from the sample were detected with a pure Ge solid-state detector through receiving slits. The angle of diffraction was $2\theta = 6°$, and the typical exposure time for each measurement was $\sim 120$ s. Each measurement was conducted in increments of $+0.3$ mm from the sample centre ($z = 0$) to the sample edge. In run #A0161, bcc iron was observed from $z = 0$ to $+0.9$, and then FeO and iron-rich olivine appeared. Subsequently, the diffraction patterns were mainly derived from olivine and iron-poor pyroxene toward the edges of the samples. This result is in good agreement with the results from SEM. The unit cell parameters of the bcc iron were $a = 2.865(1)$ Å and $V = 23.52(7)$ Å$^3$ for #A0161, and $a = 2.865(1)$ Å and $V = 23.52(5)$ Å$^3$ for #A0157, suggesting that the recovered iron under wet conditions was identical to that under dry conditions after the complete release of hydrogen.

**Data availability.** The data supporting the main findings of this study are available within the main article and its Supplementary Information (Supplementary Figs 1–4 and Supplementary Tables 1 and 2).

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

## Acknowledgements

We thank K. Funakoshi, J. Abe and S. Machida for their assistance with the neutron experiments at BL11 (PLANET), J-PARC (proposals no. 2013B0034, 2014A0049 and 2014B0044). Discussions with K. Aoki and A. Machida were helpful. This work was

supported by KAKENHI Grant-in-Aid for Scientific Research (A) (no. 25246037) from the Japan Society for the Promotion of Science (JSPS).

## Author contributions

R.I.-O., T.Y., H.G., T.O., T.H. and A.S.-F. performed the high-$P$–$T$ neutron diffraction experiments. R.I.-O., H.G. and T.Y. developed the high-pressure cell for these experiments. R.I.-O. analysed the neutron diffraction data. T.H. and A.S.-F. developed the high-pressure neutron diffractometer (PLANET). R.I.-O. and H.G. prepared the samples. T.Y. and R.I.-O. wrote the manuscript. T.Y. directed this study.

## Additional information

**Competing financial interests:** The authors declare no competing financial interests.

**Publisher's note**: 

