## [Peer Review File · Nature Communications]

Reviewers' Comments:

Reviewer #1 (Remarks to the Author)

Using ATSUHIME 6-axis press at the PLANET beamline of the J-PARC neutron facility, the authors studied the reaction between Fe and D₂O up to 5 GPa and 1000 K. They found that the Fe can be readily hydrogenated by the water. Water plus iron is a much more realistic assemblage than the previous experiment using pure hydrogen and iron. It has important implications to the Earth science and should be published in Nature Communications with considerable improvements of the presentation.

1. Nature Comm instructs reviewers to "pay particular attention to the statistics, if applicable. All error bars should be defined in the corresponding figure legends. Please include in your report a specific comment on the appropriateness of any statistical tests, and the accuracy of the description of any error bars and probability values". This ms does not give any error bar nor any statistical test. The authors should make an effort to follow the NC instructions to correct this problem.
2. The paper claims 200/111 ratio is indicative of the hydrogen contents in the fcc phase. Obviously this is based on some specific crystallographic model of the hydrogen position in FeH_x. Please provide the reference and quantitative relation.
3. How did they estimate hydrogen percentage such as 0.71 and 0.79%? Please provide reference, quantitative relation, and error bars.
4. Fig. 3 caption. "Although the data are highly scattered owing to the effect of grain growth, the ratio gradually decreased in runs #A0156 and #A0161, whereas no such tendency was observed in #A0157." The trend or no trend is by no means obvious. The author should do a linear regression and statistical test to whether there is a correlation.
5. The authors claim that "This strongly suggests that hydrogen is the first light element (silicon, sulfur, oxygen, and carbon) dissolved into iron during the Earth's evolution." However, they only studied hydrogen without comparison with other light elements. The only other element oxygen actually forms FeO before the formation of FeH_x. This claim should be deleted.
6. The first word "inevitable" in the title is inappropriate and should be deleted.

Reviewer #2 (Remarks to the Author)

Iizuka-Oku et al. present in situ neutron diffraction results at moderately high pressures and temperatures that document the formation of Fe-hydride from mixtures of Fe, SiO₂, and Mg(OH)₂. As far as I am aware, this study presents the first observations of Fe hydrides in situ at high P-T using neutron diffraction. Fe hydrides are a very important material to study, since H is a proposed candidate for Earth's core light element. They can not be easily studied by X-ray diffraction because hydrogen does not scatter X-rays well. Therefore, from a technical standpoint, this study represents a major advancement and accomplishment in our field.

The manuscript is thoughtfully written, and the figures/tables/supplementary material adequately illustrate the methodology and results (I particularly like Fig 2). The data and analyses appear to be of good quality. However, I disagree with some of the major conclusions that the authors draw from their data in terms of applications to Earth's core formation, and these conclusions are overstated. I think that the interpretation of the data would have to be significantly changed before the manuscript is publishable in any journal, but subject to these changes it could be suitable for publication in Nature Communications due to the significance of the results.

Major comments:

- The experiments were likely saturated with hydrous minerals, which the Earth was not. Saying

that you can dissolve a certain amount of H or D into metallic Fe is not the same as claiming that that much H will actually partition into the metallic phase in a chemical system that more closely resembles the Earth (e.g., low H).

- The Earth is likely to have at least partially melted (at least the Fe-alloy) fairly early in its history, due to the amount of energy deposited by impacts. Reactions with molten Fe are not the same as those with solid Fe. Once the silicate portion of the planet began to melt, H would be more likely to partition into the melt, not the solid, which is again a different phase than what was studied here. This work is really more applicable to undifferentiated asteroids that formed late enough that ^{26}Al was dead but before impacts were large enough to cause melting. This could still be applicable to the Earth, if it occurred in the building blocks of the Earth and didn't reverse upon core formation. But as you state, the solubility of H in Fe is very low at ambient P; it is unclear if this mechanism would work at the very low pressures and temperatures of an asteroid interior.
- The mechanism that you propose only works very early in a planet/planetesimal's history; this assumes that H is already present very early. This is much more likely in the outer Solar System, which is thought to have been more volatile-rich, than where the terrestrial planets were forming. N-body simulations show that Earth accretes material from farther from the Sun (more volatile rich) toward the end of its accretion, when melting was extensive and impacts may have been energetic enough to cause significant volatile loss.
- Because of these issues, I think that the claim that "hydrogenation of iron occurs inevitably" (and the title of the paper) is greatly exaggerated.

Minor comments:

- If you have any patterns in which the peaks from the silicates/oxides are visible, you should provide an example with all of those peaks identified, like you did in extended data figure 4, to illustrate that the expected reaction actually occurred.
- You need to provide all of the data that you analyzed in extended data table 3, not just results from one pattern.
- You need to report some uncertainties on D contents.
- To make Fig 3 more convincing, maybe you could fit the data from each of the runs to a line and label the figure with the slopes (and their uncertainties).
- Figure 4: you should make the scale bars more legible, and mark where the enlarged image comes from. Why is there so much O in the Fe metal?
- Eq 1 is not balanced.
- Would the results change if the experiments contained H instead of D?

Reviewer #3 (Remarks to the Author)

Dear Editor,

The paper by Iizuka-Oku et al. on the hydrogenation of iron presents important experimental results concerning the chemical reaction between solid iron and solid hydrous phases. The authors performed a series of experiments in a multi-anvil press up to a few GPa and 1000K temperature. The experiments show beyond any doubt that a chemical reaction occurs, which leads to the (partial) decomposition of the hydrated phase and partial migration of H into the iron. This result

suggests that a certain amount of H could be transferred into iron in the first steps of the core formation. As such, the paper is important and brings new data to the long-standing debate about the place of H in the Earth's core. I think the paper is probably publishable, but I suggest a major revision of the interpretation of these experimental data.

Iron in this experiments clearly acts as a catalyst to trigger the reaction of H decomposition. This is well known and widely used in chemical engineering. I think a comment about the catalytic effect and reference, for example to a review paper, would be welcome here. It is also true that those catalytic reactions were not followed up to high pressures and temperatures, and thus this study brings indeed important insight in this process. This could also be better underlined in the paper.

Concerning the implications of this study to the Earth debate, the experiments are limited for obvious reasons to 5 GPa and 1000K. There is no indication whatsoever that once in the solid structure hydrogen would actually stay there at higher pressures and temperatures. For example these experiments do not even touch the stability field of the hcp phase - the dominant phase in the Earth's core. There is no indication about the partitioning of hydrogen and its solubility in the hcp phase. Maybe the solubility drops and H would be exsolved in a natural setting.

Moreover they also come short of the melting line. The process of core formation involved melts, both silicate and metallic and this is an extremely important point. As the metallic melts percolate downwards through the magma ocean they will react and, at least partially, equilibrate with the silicate melts. This experiments on solid phases offer no clue about the behavior of the melts. Maybe all H would partition to the silicate fraction.

Recent first-principles calculations, completely ignored by the authors here, suggest that from a seismological point of view H could be primary a light element in the outer core (Umemoto and Hirose, 2015), but not in the inner core (Caracas, 2015). On the other hand isotope measurements suggest that H is not a major element in the core (Shahar et al, 2016).

It seems that the evidence actually points towards a hydrogen-free core. Even if the debate is not yet finished, these experiments address only a limited region of the phase space, and the extrapolations and implications the authors claim seem to be unsupported.

12th. September 2016

To Reviewers:

First of all, we deeply appreciate for all the three reviewers who recognized the importance of our work and gave us many constructive comments. They are really helpful to improve the manuscript. We agreed most of the comments, but there are also some misunderstandings, which were caused by the improper descriptions of our scenario in the original draft. We revised many parts in the whole manuscript accordingly and reply to each comment point by point in the following sections.

Reviewers' comments:

Reviewer #1 (Remarks to the Author): Using ATSUHIME 6-axis press at the PLANET beamline of the J-PARC neutron facility, the authors studied the reaction between Fe and D₂O up to 5 GPa and 1000 K. They found that the Fe can be readily hydrogenated by the water. Water plus iron is a much more realistic assemblage than the previous experiment using pure hydrogen and iron. It has important implications to the Earth science and should be published in Nature Communications with considerable improvements of the presentation.

1. Nature Comm instructs reviewers to "pay particular attention to the statistics, if applicable. All error bars should be defined in the corresponding figure legends. Please include in your report a specific comment on the appropriateness of any statistical tests, and the accuracy of the description of any error bars and probability values". This ms does not give any error bar nor any statistical test. The authors should make an effort to follow the NC instructions to correct this problem.

- ✓ We added errors for each value in the corresponding tables and the manuscript. Error bars were given in figures or relevant descriptions on the accuracy were added in the captions.

2. The paper claims 200/III ratio is indicative of the hydrogen contents in the fcc phase. Obviously this is based on some specific crystallographic model of the hydrogen position in FeH_x. Please provide the reference and quantitative relation.

- ✓ We calculated using the same methods as Machida *et al* (2014) (Ref. 28), in which the D atoms occupy both O and T sites of fcc meta lattice. The details of

the quantitative relation are described in their papers and the reference number was give clearly in the main text.

3. *How did they estimate hydrogen percentage such as 0.71 and 0.79%? Please provide reference, quantitative relation, and error bars.*

- ✓ Deuteration contents x were calculated by using the following relation, $x = \frac{V(\text{FeD}_x) - V(\text{Fe})}{\Delta V_D}$, where the value of ΔV_D (the volume increase per deuterium atom) was taken from Ref. 28. This equation, the reference, and the errors were added in the main text.

4. *Fig. 3 caption. "Although the data are highly scattered owing to the effect of grain growth, the ratio gradually decreased in runs #A0156 and #A0161, whereas no such tendency was observed in #A0157." The trend or no trend is by no means obvious. The author should do a linear regression and statistical test to whether there is a correlation.*

- ✓ The linear fitting ($y = A - Xt$) was made for all the data and the argument was added in the manuscript using the value X and the coefficient of determination (R^2). Because of the large scatter of the data, this intensity data alone is not strong enough but the important point is that intensity data is in harmony with the clear data obtained from the unit cell volume. The main text and Figure 3 were revised accordingly.

5. *The authors claim that "This strongly suggests that hydrogen is the first light element (silicon, sulfur, oxygen, and carbon) dissolved into iron during the Earth's evolution." However, they only studied hydrogen without comparison with other light elements. The only other element oxygen actually forms FeO before the formation of FeH_x. This claim should be deleted.*

- ✓ All other light elements dissolve into **molten** iron (Ref. 2), while we found that hydrogen dissolves into **solid** iron at much lower temperatures. From this fact, we can say that "hydrogen is the first light element" without doing additional experiments. Most of the FeO formed with FeH_x reacts with silicates. We modified the main text and removed the word "strongly".

6. *The first word "inevitable" in the title is inappropriate and should be deleted.*

✓ We deleted the word "inevitable" from the title.

Reviewer #2 (Remarks to the Author): Iizuka-Oku et al. present in situ neutron diffraction results at moderately high pressures and temperatures that document the formation of Fe-hydride from mixtures of Fe, SiO₂, and Mg(OD)₂. As far as I am aware, this study presents the first observations of Fe hydrides in situ at high P-T using neutron diffraction. Fe hydrides are a very important material to study, since H is a proposed candidate for Earth's core light element. They cannot be easily studied by X-ray diffraction because hydrogen does not scatter X-rays well. Therefore, from a technical standpoint, this study represents a major advancement and accomplishment in our field.

The manuscript is thoughtfully written, and the figures/tables/supplementary materials adequately illustrate the methodology and results (I particularly like Fig 2). The data and analyses appear to be of good quality. However, I disagree with some of the major conclusions that the authors draw from their data in terms of applications to Earth's core formation, and these conclusions are overstated. I think that the interpretation of the data would have to be significantly changed before the manuscript is publishable in any journal, but subject to these changes it could be suitable for publication in Nature Communications due to the significance of the results.

Major comments:

- The experiments were likely saturated with hydrous minerals, which the Earth was not. Saying that you can dissolve a certain amount of H or D into metallic Fe is not the same as claiming that that much H will actually partition into the metallic phase in a chemical system that more closely resembles the Earth (e.g., low H).

✓ Starting material in the present study is the same or very similar to those used in the previous works (Refs. 15,19,23), which resembled the primordial material. The amount of water content in the primordial material is still controversial and the present starting material is one of the well-accepted model materials. Important point in the present work is to show quantitatively that a considerable amount of hydrogen dissolves into iron if the hydrous mineral exists. Some words and references were added in the main text.

- The Earth is likely to have at least partially melted (at least the Fe-alloy) fairly early in its history, due to the amount of energy deposited by impacts. Reactions with molten Fe are not the same as those with solid Fe. Once the silicate portion of the planet began to melt, H would be more likely to partition into the melt, not the solid, which is again a different phase than what was studied here. This work is really more applicable to undifferentiated asteroids that formed late enough that ^{26}Al was dead but before impacts were large enough to cause melting. This could still be applicable to the Earth, if it occurred in the building blocks of the Earth and didn't reverse upon core formation. But as you state, the solubility of H in Fe is very low at ambient P; it is unclear if this mechanism would work at the very low pressures and temperatures of an asteroid interior.

- ✓ Our study simulates the stage of Earth formation that the mass of the proto Earth became more than about 1 % of the present Earth and its interior became above 3 GPa. If the temperature inside of such proto Earth rose above the dissociation temperature of hydrous minerals, hydrogenation of iron occurred while iron remains solid. This causes the decrease of melting temperature of iron more than 500K. Then, the iron hydride melts first and molten iron hydride will sink to the deep part easily. Even though the giant impact occurred, it is likely, although not certain, that many parts of the Earth material were subjected to above 3 GPa and a large part of hydrogen were retained. Main text was revised to explain this scenario clearly.

- The mechanism that you propose only works very early in a planet/planetesimal's history; this assumes that H is already present very early. This is much more likely in the outer Solar System, which is thought to have been more volatile-rich, than where the terrestrial planets were forming. N-body simulations show that Earth accretes material from farther from the Sun (more volatile rich) toward the end of its accretion, when melting was extensive and impacts may have been energetic enough to cause significant volatile loss.

- ✓ Thank you for a good comment. Unfortunately, however, it is out of the scope of the present paper. Since we changed the main text as described above, the

assumed condition became clearer and some readers may use our results for their own discussions.

- Because of these issues, I think that the claim that "hydrogenation of iron occurs inevitably" (and the title of the paper) is greatly exaggerated.

- ✓ We deleted the words "inevitable" and "inevitably" from the title and the manuscript.

Minor comments:

- If you have any patterns in which the peaks from the silicates/oxides are visible, you should provide an example with all of those peaks identified, like you did in extended data figure 4, to illustrate that the expected reaction actually occurred.

- ✓ We added marks for all the visible peaks from the silicates/oxides, together with those from NaCl pressure marker, in the patterns in Figure 1 and Supplementary Figure 3.

- You need to provide all of the data that you analyzed in extended data table 3, not just results from one pattern.

- ✓ We provide all of the data analyzed in revised Supplementary Table 2.

- You need to report some uncertainties on D contents.

- ✓ We added uncertainties of D contents in the text and tables.

- To make Fig 3 more convincing, maybe you could fit the data from each of the runs to a line and label the figure with the slopes (and their uncertainties).

- ✓ Linear fittings were made and slopes were added in Fig. 3. Discussion using uncertainty (R^2 : coefficient of determination) was added in the main text.

- Figure 4: you should make the scale bars more legible, and mark where the enlarged image comes from. Why is there so much O in the Fe metal?

- ✓ Figure 4 was revised as suggested. The reason why much oxygen exists in iron is apparently due to rapid oxidization of Fe after polishing the cross section of the recovered sample.

- Eq 1 is not balanced.

- ✓ The Eq.1 was corrected as “ $2\text{Fe} + \text{D}_2\text{O} \rightarrow \text{FeO} + \text{FeD}_x + (1-x/2)\text{D}_2 \uparrow$ ”.

- Would the results change if the experiments contained H instead of D?

- ✓ Our experimental results are consistent with the results of previous works (e.g., Refs. 8, 9, 10), in which H was used instead of D and we could not find any isotope effect. Short description was added in the “Methods”.

Reviewer #3 (Remarks to the Author):

Dear Editor,

The paper by Iizuka-Oku et al. on the hydrogenation of iron presents important experimental results concerning the chemical reaction between solid iron and solid hydrous phases. The authors performed a series of experiments in a multi-anvil press up to a few GPa and 1000K temperature. The experiments show beyond any doubt that a chemical reaction occurs, which leads to the (partial) decomposition of the hydrated phase and partial migration of H into the iron. This result suggests that a certain amount of H could be transferred into iron in the first steps of the core formation. As such, the paper is important and brings new data to the long-standing debate about the place of H in the Earth's core. I think the paper is probably publishable, but I suggest a major revision of the interpretation of these experimental data.

Iron in these experiments clearly acts as a catalyst to trigger the reaction of H decomposition. This is well known and widely used in chemical engineering. I think a comment about the catalytic effect and reference, for example to a review paper, would be welcome here. It is also true that those catalytic reactions were not followed up to high pressures and temperatures, and thus this study brings indeed important insight in this process. This could also be better underlined in the paper.

- ✓ We appreciate the comment. In the present system, however, iron has changed after the reaction and we do not think the word “catalyst” is proper in this case. We added description that iron triggers the decomposition of H_2O in the main text.

Concerning the implications of this study to the Earth debate, the experiments are limited for obvious reasons to 5 GPa and 1000K. There is no indication whatsoever that once in the solid structure hydrogen would actually stay there at higher pressures and temperatures. For example these experiments do not even touch the stability field of the hcp phase - the dominant phase in the Earth's core. There is no indication about the partitioning of hydrogen and its solubility in the hcp phase. Maybe the solubility drops and H would be exsolved in a natural setting.

- ✓ Our scenario is that the hydrogenation of iron occurs at low pressures before the iron changes into *hcp* phase. Hydrogenation drops the melting temperature of iron more than 500K and molten iron hydride was carried to the deep part in the proto Earth. There are many arguments about the distribution of hydrogen between molten outer core and solid inner core with *hcp* structure (e.g., ref. 17), but it is beyond the scope of the present work. The main text was revised so that this scenario could become clearer.

Moreover they also come short of the melting line. The process of core formation involved melts, both silicate and metallic and this is an extremely important point. As the metallic melts percolate downwards through the magma ocean they will react and, at least partially, equilibrate with the silicate melts. These experiments on solid phases offer no clue about the behavior of the melts. Maybe all H would partition to the silicate fraction.

- ✓ We agree that the melting of iron and silicates are very important in the core formation. The main point in this paper is that the hydrogenation of iron occurs even when the iron remain solid. Behavior at higher temperature has been studied by quench experiments (e.g., Refs. 9, 15), which clearly showed that once iron is hydrogenated it melts before the silicates. As stated in the main text, it will become important to study the reaction between the molten iron hydride and solid or partially molten silicates.

Recent first-principles calculations, completely ignored by the authors here, suggest that from a seismological point of view H could be primary a light element in the outer core (Umemoto and Hirose, 2015), but not in the inner core (Caracas, 2015). On the

other hand isotope measurements suggest that H is not a major element in the core (Shahar et al, 2016).

- ✓ References above were added in the introduction.

It seems that the evidence actually points towards a hydrogen-free core. Even if the debate is not yet finished, these experiments address only a limited region of the phase space, and the extrapolations and implications the authors claim seem to be unsupported.

- ✓ We agree that our experiments address only very limited region of the “light elements problem”. All we can do is to show some possibilities based on the solid experimental results. We modified the main text and softened the expressions so that our scenario becomes clear.

Reviewers' Comments:

Reviewer #1 (Remarks to the Author)

I am satisfy with the authors' reply and the revised manuscript and recommend the paper to be published in Nature Communications.

Reviewer #2 (Remarks to the Author)

Iizuka-Oku et al. report on the deuteration of solid iron in contact with deuterated silicates using in situ neutron diffraction at high P-T. I previously reviewed this manuscript as Reviewer #2. As I mentioned before, I think the data presented here are great, and this study is very timely and important. Therefore I think it is appropriate to publish in Nature Communications. I am generally very happy with the revisions and responses that the authors have made to my comments and those of the other reviewers, and I feel that the paper is significantly improved as a result of the revisions that have been undertaken. I only have two lingering issues that I think will be very easy to address; following these minor changes, I think the paper is ready to be published in this journal.

First, this mechanism is only applicable to the earliest stages of Earth's formation, while everything is solid. You can not claim, based on this, that "therefore, it is likely that hydrogen is the primary light element in the core" (L199-200). This sentence needs to be removed or edited to say something like "therefore, it is possible that hydrogen was the primarily light element in the core very early in Earth's history". This early material added to the core might have contained ~1 wt% of H as you argue, but this accounts for a very small fraction of the core. Most core material would have experienced liquid metal - liquid silicate reactions, which aren't addressed here, so it remains unknown how much H this later material (that makes up most of the core) could have contained.

Further, in order for H to be the first light element in the core, it is required that the earliest material that went into forming the Earth was volatile rich material, which you say only makes up ~10% of the bulk Earth (L192). This assumption needs to be clearly stated, because it is a very particular scenario.

I also want to compliment the authors on the mention of methods that were unsuccessful (L249-250); this is done far too infrequently in our field, and is very helpful for other researchers.

22nd. November 2016

To #2 Reviewer:

We deeply appreciate for the reviewers, who recognized the importance of our work and gave us many constructive comments. They are really helpful to improve the manuscript. We agreed most of the comments, but there is also a misreading. We revised the parts in the manuscript accordingly and reply to each comment from #2 reviewer point by point below.

#2 Reviewer's comments:

1. *First, this mechanism is only applicable to the earliest stages of Earth's formation, while everything is solid. You cannot claim, based on this, that "therefore, it is likely that hydrogen is the primary light element in the core" (L199-200). This sentence needs to be removed or edited to say something like "therefore, it is possible that hydrogen was the primarily light element in the core very early in Earth's history". This early material added to the core might have contained ~1 wt% of H as you argue, but this accounts for a very small fraction of the core. Most core material would have experienced liquid metal - liquid silicate reactions, which aren't addressed here, so it remains unknown how much H this later material (that makes up most of the core) could have contained.*

- ✓ We changed the sentence as suggested, but added short sentence to make the connection to the continued sentence smooth.

2. *Further, in order for H to be the first light element in the core, it is required that the earliest material that went into forming the Earth was volatile rich material, which you say only makes up ~10% of the bulk Earth (L192). This assumption needs to be clearly stated, because it is a very particular scenario.*

- ✓ I think this comment is a misreading of the reviewer. The mixing ratio of low and high temperature condensates has nothing to do with the hydrogen to be the first light element in the core. The sentence from L192 to L194 says that, in order to explain the 10% density deficit of the present core, only 10% volatile rich material is enough in the starting material.

3. *I also want to compliment the authors on the mention of methods that were unsuccessful (L249-250); this is done far too infrequently in our field, and is very helpful for other researchers.*

- ✓ We revised the sentences concerned in the methods by adding more detailed information about results and reasons for our unsuccessful experiments.